# A New Social Media-Driven Cyber Threat Intelligence

Fahim Sufi 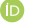

School of Public Health and Preventive Medicine, Monash University, Melbourne, VIC 3004, Australia; fahim.sufi@monash.edu

**Abstract:** Cyber threats are projected to cause USD 10.5 trillion in damage to the global economy in 2025. Comprehending the level of threat is core to adjusting cyber posture at the personal, organizational, and national levels. However, representing the threat level with a single score is a daunting task if the scores are generated from big and complex data sources such as social media. This paper harnesses the modern technological advancements in artificial intelligence (AI) and natural language processing (NLP) to comprehend the contextual information of social media posts related to cyber-attacks and electronic warfare. Then, using keyword-based index generation techniques, a single index is generated at the country level. Utilizing a convolutional neural network (CNN), the innovative process automatically detects any anomalies within the countrywide threat index and explains the root causes. The entire process was validated with live Twitter feeds from 14 October 2022 to 27 December 2022. During these 75 days, AI-based language detection, translation, and sentiment analysis comprehended 15,983 tweets in 47 different languages (while most of the existing works only work in one language). Finally, 75 daily cyber threat indexes with anomalies were generated for China, Australia, Russia, Ukraine, Iran, and India. Using this intelligence, strategic decision makers can adjust their cyber preparedness for mitigating the detrimental damages afflicted by cyber criminals.

**Keywords:** electronic warfare threat; cyber threat index; artificial intelligence in cyber; anomaly detection; convolutional neural network; deep learning; social media-driven intelligent system

## 1. Introduction

Cyber-attacks and threats caused about USD 1 trillion in damage to the global economy in 2020 [1]. This threat is projected to grow more than tenfold by 2025, with 10.5 trillion (in USD) in damage to the global economy [2]. To mitigate the global economic loss of a cyber-related threat, timely and effective analysis of cyber-related data using deep learning (DL) is required. Previous research has demonstrated the use of the DL technique on network traffic data for generating cyber situational awareness [3,4]. DL techniques such as anomaly detection were also used on simulated sensor data, as shown in [5]. Other than analyzing network traffic data or simulated sensor data, primary data collected through surveys or questionnaires can also provide intelligence on cyber threats [6]. However, cyber threat intelligence derived from critical DL-based analysis of social media data has never been demonstrated within existing studies [3–9].

Recent research works on social media-based analysis have demonstrated dynamic creation of threat indexes using keyword-based extraction [10,11]. However, research shown in [10,11] provided threat intelligence on different dimensions of COVID-19. While study [10]. did not use any DL techniques, DL was used in [11]. This paper used keyword-based extraction of targeted social media information for generating cyber threat indexes and used DL-based anomaly detection on the time-series index (similar to techniques demonstrated in [11]). Available cyber-related studies such as [12,13] only provide cyber defence capability on a personal or organizational level.

According to the literature and to best of my knowledge, existing studies do not report DL-based analytical capability on global countrywide (i.e., national level) cyber threats.

However, this paper reports the first study that generates and analyzes time-series indexes on cyber threat analysis. Time-series-based analysis of cyber threat intelligence (depicted in this study) provides in-depth countrywide cyber threat awareness. Using countrywide cyber threat intelligence, strategic decision makers of any country can adjust their cyber preparedness to mitigate the detrimental damages afflicted by cyber criminals. The ability to effectively defend against or launch cyber threats can provide a significant advantage on the battlefield as well.

The innovative methods described within this study were tested and evaluated on a live Twitter feed from 14 October 2022 to 27 December 2022. During these 75 days, global social media posts related to cyber were captured and analyzed with artificial intelligence (AI)-based services such as language detection, translation, and sentiment analysis. By performing data manipulations on 15,983 tweets from 15,315 users (in 47 different languages), cyber threat index for 6 different countries were produced. Finally, convolutional neural network (CNN)-based anomaly detection (a deep learning method) was used to automatically identify anomalies on cyber threat indexes for China, Australia, Russia, Ukraine, Iran, and India. Unlike [14–16], this work was evaluated on a mobile environment. It should be mentioned that these countries were not selected in a random manner. They were selected by considering a range of factors such as the *number of social media users* (i.e., the US and India have a huge number of Twitter users [17]), the *concerns of the social media users* (i.e., US Twitter users perceiving China and Russia as a cyber threat [18,19]), and *current issues* (e.g., Russia-Ukraine cyber war, cyber-attacks in Australia [20–23] etc.).

CNN was chosen as opposed to other deep learning methods such as the recurrent neural network (RNN), since CNN have reportedly been used in mobile platforms for solving different research problems [16]. RNN, long short-term memory networks (LSTMs), and other DL implementations are not supported on low-code mobile platforms [16].

## 2. Materials and Methods

Several literatures on social media-driven cyber intelligence were reviewed to find the disadvantages and drawbacks. With the help of a systematic literature review, the five disadvantages of social media-driven cyber intelligence were identified. These disadvantages were translated as system requirements, followed by the assignment of appropriate solution components to these requirements. After developing the solution architecture, the system was developed. Thus, the system development framework stipulated by The Open Group Architectural Framework (TOGAF) and Archimate standard were broadly followed [24], as seen from Figure 1. Finally, the system was tested from 11 October 2022 to 27 December 2022 using a live Twitter Feed.

### 2.1. Systematic Literature Review

Keywords such as "cyber threat intelligence", "social media", etc. were used from peer-reviewed databases such as IEEE Explore, Scopus, ACM Library, Web of Science, and others. About 76 articles were identified from these peer-reviewed sources. Search engines such as Google and Bing also resulted in 23 grey literatures. Out of these 99 literatures, about 30 literatures were identified discussing the challenges of social media-driven cyber solutions. Table 1 shows the details of inclusion and exclusion for both peer-reviewed and grey literature searches.

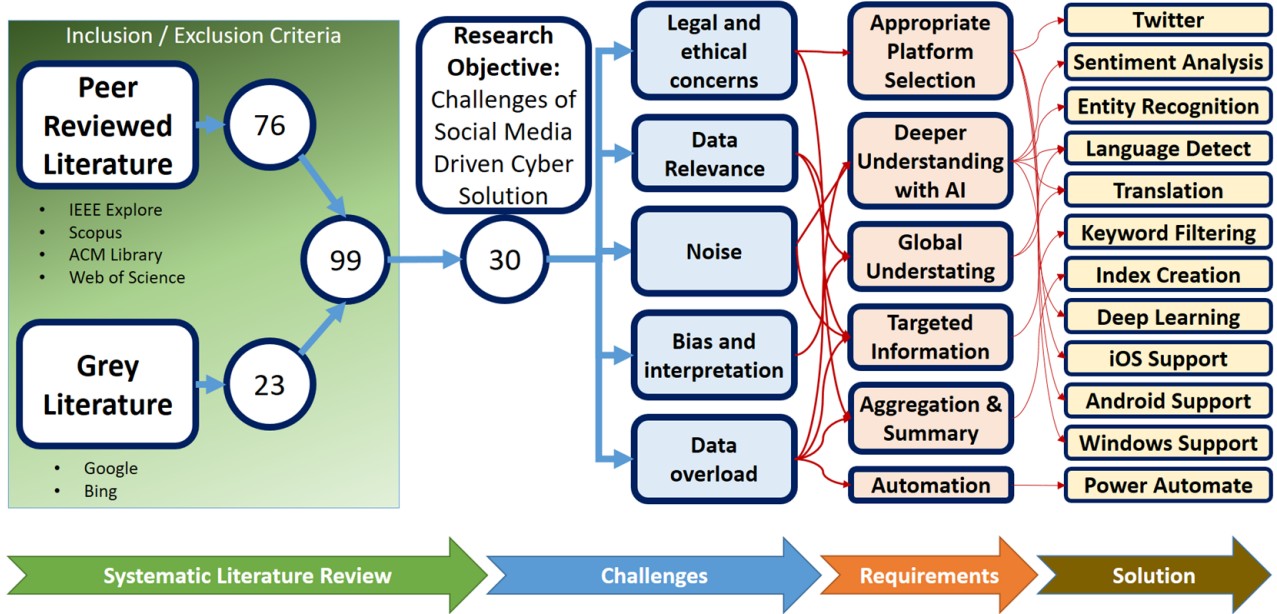

**Figure 1.** Overview of the design for a new cyber index system by a systematic literature review, finding challenges, developing requirements, and finally, designing the solution.

**Table 1.** Inclusion and exclusion criteria for both peer-reviewed and grey literature.

| Category | | Criteria |
|---|---|---|
| Inclusion | Peer-Reviewed Literature | • Search Criteria used:<br>  1. "Cyber threat intelligence" + "social media"<br>  2. "Cyber intelligence" + "social media data"<br>  3. "Cyber threat" + "social media"<br>• Review studies, survey/questionnaire-based qualitative or quantitative studies, original research articles<br>• Papers indexed in popular peer-reviewed sources (i.e., IEEE Explore, ACM Library, Scopus, and Web of Science)<br>• Papers focusing on research questions (i.e., challenges on social media-based cyber intelligence)<br>• Studies available in English<br>• Studies available in full text |
| | Grey Literature | • Websites focused on social media-based cyber intelligence<br>• Indexed in popular search engines (i.e., Google and Microsoft Bing)<br>• Articles authored by either by a cyber solution vendor or a third-party bench marking company<br>• Articles available in English |
| Exclusion | Peer-Reviewed Literature | • Tutorial Papers<br>• Short papers less than four pages<br>• Poster papers, editorials, abstract (i.e., lacking detailed Information) |
| | Grey Literature | • Websites referring to peer-reviewed literature<br>• Cyber solutions promoted by bloggers, consultants, or third party companies<br>• Tutorial videos and discussions on cyber solutions |

## 2.2. Challenges of Social Media-Driven Cyber Solutions

Social media-based cyber intelligence, also known as social media intelligence (SOCMINT), can provide valuable insights into online activity and help identify potential threats. However, there are some drawbacks and challenges associated with this type of intelligence, as discussed in [25–51]. Table 2 shows how the five major challenges were obtained through a literature review. The five major challenges are discussed briefly in this section.

**Table 2.** Five critical challenges of social media-driven cyber solutions.

| | [25] | [26] | [27] | [28] | [29] | [30] | [31] | [32] | [33] | [34] | [35] | [36] | [37] | [38] | [39] | [40] | [41] | [42] | [43] | [44] | [45] | [46] | [47] | [48] | [49] | [50] | [51] |
|---|---|---|---|---|---|---|---|---|---|---|---|---|---|---|---|---|---|---|---|---|---|---|---|---|---|---|---|
| Legal /Ethics Concern | | | | • | | | | | | | | • | | | | | | • | | | | | | • | | | |
| Data Relevance | • | | | | • | | | | • | | | | | | | | | | • | | | | | | • | | |
| Noise | | | • | | | • | | | | • | | | | • | • | | | | | | • | | | | | • | • |
| Bias and Interpretation | • | | | | | • | | | | | | | | | | | • | | | • | | • | | | | | |
| Data Overload | | | | • | | | • | • | | | • | | • | | | • | | | | | • | | | | | | |

### 2.2.1. Legal and Ethical Concerns

The collection and use of data from social media can raise legal and ethical concerns. There may be questions around privacy and the legality of collecting certain types of information, and there may also be issues around how the data are stored and used.

### 2.2.2. Data Relevance

Social media networks create much data, but not all of it is equally insightful or useful for cyber threat intelligence. It may be challenging to distinguish the data that is most pertinent and valuable from information that is not directly related to cyber threats, such as personal beliefs, political viewpoints, or jokes.

### 2.2.3. Noise

Noise is the enormous amount of data and information produced on social media platforms, which makes it difficult to determine the pertinent and usable information for intelligence purposes. Social media sites produce a vast amount of data, including text, photos, videos, and other sorts of media, due to the daily uploading of content by millions of users.

### 2.2.4. Bias and Interpretation

The interpretation of social media data can be influenced by the biases of the analyst. These biases may be based on race, religion, culture, ethnicity, or even gender. For example, a person living in the US might have a preconception that most of the cyber-attacks are of Russian, Chinese, or North Korean origin. People of Iran may think that the US conducts most of the cyber-attack to create religious tensions within Middle East. Different analysts may interpret the same social media post in different ways, leading to inconsistent results.

### 2.2.5. Data Overload

Social media generates massive amounts of data, which can be overwhelming to process and analyze. This can result in missed or overlooked data that may be important in identifying potential threats or risks.

Overall, social media-based cyber intelligence can be a useful tool for identifying potential threats and risks. However, it is important to be aware of the limitations and potential drawbacks, and to use this type of intelligence in a responsible and ethical manner.

### 2.3. Requirement of a New Cyber Threat Index System

As seen from Figure 2, the requirements for the cyber threat index are generated from the critical analysis of the challenges. Once the core challenges of social media-driven cyber intelligence solutions were identified, six requirements for the proposed cyber threat index system were identified. The following are the brief descriptions of these requirements (as they appear in Table 3).

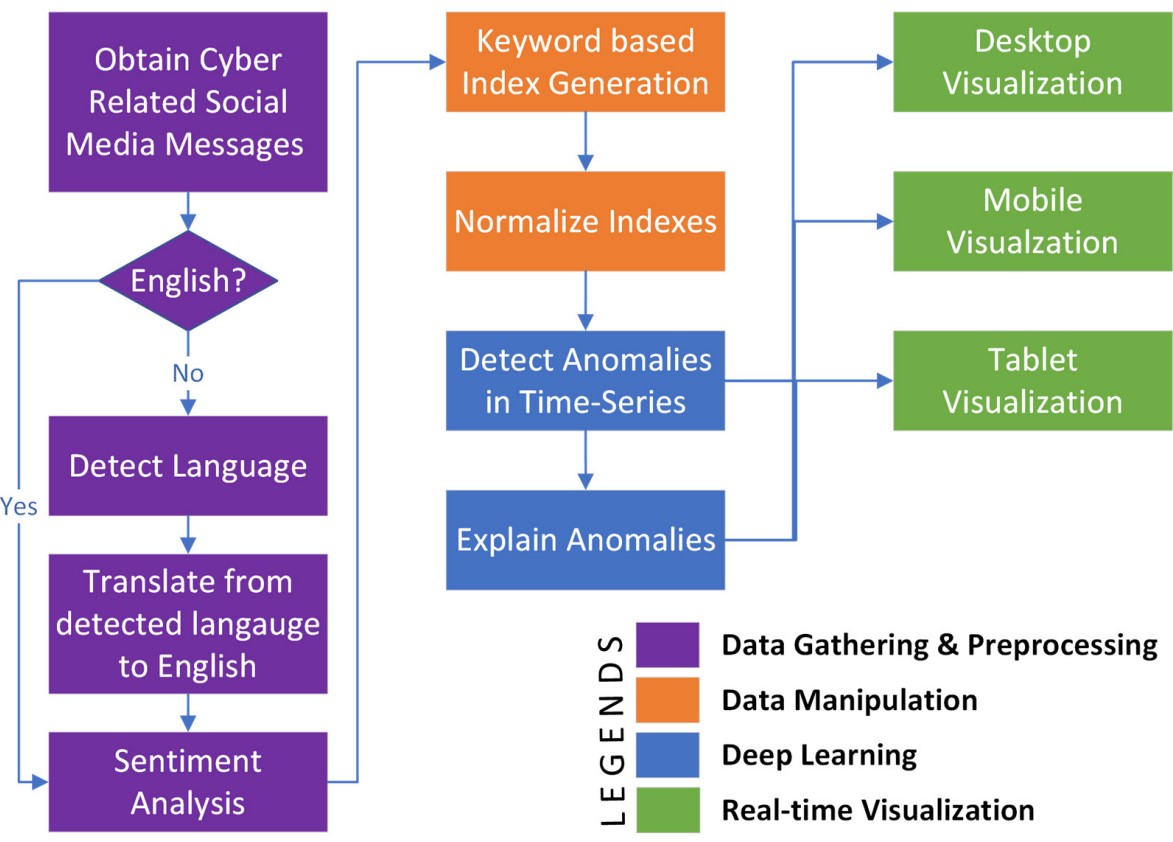

**Figure 2.** Conceptual diagram of deep learning-based, countrywide cyber index implementation.

**Table 3.** Mapping between requirements and challenges.

|  | Legal and Ethical Concerns | Data Relevance | Noise | Bias and Interpretation | Data Overload |
|---|:---:|:---:|:---:|:---:|:---:|
| Appropriate Platform Selection | ● |  |  |  |  |
| Deeper Understanding with AI |  |  | ● |  | ● |
| Global Understanding |  | ● |  | ● |  |
| Targeted Information |  | ● | ● |  | ● |
| Aggregation & Summary | ● | ● |  |  | ● |
| Automation |  |  |  |  | ● |

### 2.3.1. Appropriate Platform Selection

Not all the social media platforms are exempt from ethical concerns. There are ethical issues involved in closed network-based social media platforms such as Snapchat and Facebook, since the user of a closed network believes that the shared information is only available within the social media user's approved connections [52,53]. However, for open network-based social media platforms such as Twitter, Pinterest, and Instagram, the user already knows that all shared information is available at the public level. Hence, for open network-based social media platforms, there are limited legal and ethical concerns.

### 2.3.2. Deeper Understanding with AI

Usage of AI, ML, and DL provides deeper insights on the vast social media corrupted with noise and data errors. With modern natural language processing (NLP) techniques, AI allows modern systems to handle issues such as "data overload" and "noise". This paper applies NLP-based algorithms to generate a deeper understanding of cyber threats.

### 2.3.3. Global Understanding

Social media posts contain much irrelevant information filled with biases. What is often referred to as an issue might be completely irrelevant in another country. To correctly interpret the data with the proper context, a desired system needs to have a global understanding of global locations, languages, and cultures.

### 2.3.4. Targeted Information

Since social media data is overwhelmed with irrelevant data mixed with noise, it is crucial for a researcher to obtain appropriate data. Data could be appropriately targeted with keyword-based filtering to remove all unwanted information along with noise. Moreover, keyword-based filtering allows researchers to only work with a limited (as well as relevant) dataset, as opposed to dealing with big data.

### 2.3.5. Aggregation and Summary

To overcome the ethical and legal implications of using social media data for gathering intelligence, existing researchers have suggested using aggregated data [52,53]. By aggregating and summarizing all the personal information of the social media user, it becomes obfuscated. Moreover, with aggregation, the researchers no longer have to deal with huge data.

### 2.3.6. Automation

Since social media is enormous, it is often impossible to manually analyze the information. Using AI-driven automation technology, the huge data coming from social media could be processed and analyzed with computing facilities.

Table 3 demonstrates how these requirements (e.g., appropriate platform selection, deeper understanding with AI, global understanding, targeted information, aggregation and summary, and automation) satisfy each of the challenges incurred by using social media for cyber intelligence.

### 2.4. Design of a New Social Media-Driven Cyber Threat Index System

From the requirements stipulated in Section 2.3, Table 4 shows 12 solution blocks for designing the new social media-driven ETW index system. First, Twitter was selected as the source of information since Twitter is an open platform with limited ethical concerns. As a result, Twitter was previously used for designing innovative intelligence gathering systems in [11,54–63]. AI and natural language processing (NLP)-based sentiment analysis, entity recognition, language detect, translation, and deep learning provide a deeper understanding of social media messages. Both language detection and translation provide global understanding. Next, keyword-based filtering provides a means of acquiring targeted information (as previously discussed in Section 2.3.4.). Most importantly, by creating an

index (i.e., a single numerical score), millions of social media posts could be summarized. Hence, index creation facilitates aggregation and summary. Microsoft's Power Platform was used for automation [64]. Selection of this AI-driven low-code platform (i.e., Microsoft Power Platform) also facilitated interactive intelligent dashboards in iOS, Android, and Windows environments [65].

**Table 4.** Mapping between solutions and requirements.

| | Appropriate Platform Selection | Deeper Understanding with AI | Global Understanding | Targeted Information | Aggregation and Summary | Automation |
|---|---|---|---|---|---|---|
| Twitter | ● | | | | | |
| Sentiment analysis | | ● | | | | |
| Entity recognition | | ● | | | | |
| Language detection | | ● | ● | | | |
| Translation | | ● | ● | | | |
| Keyword filtering | | | | ● | | |
| Index creation | | | | | ● | |
| Deep learning | | ● | | | | |
| iOS support | ● | | | | | |
| Android support | ● | | | | | |
| Windows support | ● | | | | | |
| Microsoft Power Automate | | | | | | ● |

*2.5. Implementation and Evaluation of the Solution*

From a high-level perspective, the deep learning-based, countrywide cyber threat analysis method is composed of the following four stages:

- Data gathering and preprocessing;
- Data manipulation;
- Deep learning;
- Real-time visualization.

As seen from Figure 1, each of these stages are composed of sub-stages, which are described within this section.

2.5.1. Data Gathering and Preprocessing

Within the scope of this paper, cyber-related data were aggregated from live social media feeds (e.g., Twitter) using web scrapping and application programming interfaces (APIs). Twitter APIs and other third-party tools allow dynamic collection of targeting social media messages for further processing. In [11], keywords such as "COVID" and "corona" were used to dynamically obtain COVID-19 related social media messages. Similarly, a list of 10 keywords such as "earthquake", "flood", "bushfire", "landslide", "landfall", "tornado", "volcano", etc., were used in [58,59] to obtain social media messages related

to disasters. Similarly, "cyber" was used as keyword to obtain cyber-related social media messages for the method described within this research.

Once targeted messages are obtained, a range of artificial intelligence (AI)-based algorithms are executed as data preparation and preprocessing. These AI-based data preprocessing algorithms include language detection, translation, and sentiment analysis as demonstrated in [11,58–62]. As seen from Figure 1, for all non-English social media messages related to cyber, the language is detected followed by translation to English. Microsoft Cognitive Services' Text Analytics API [66] is used for performing this AI-driven task of language detection and dynamic translation. Microsoft Cognitive Services' Text Analytics API [66] is also used for performing sentiment analysis on the English-translated text (since sentiment analysis is only supported in the English language). Thus, by employing language detection and translation before the sentiment analysis process, social media posts in almost 110 different languages could be comprehended, as demonstrated in [11,58–62]. Other than comprehending social media posts and messages, this technique could also gather and analyze online news and articles from thousands of sources such as websites, online newspapers, blogs, etc. For example, global events from 2397 different global news sources have recently been obtained and analyzed in [60–62].

### 2.5.2. Data Manipulation

Within this stage, cyber threat indexes are generated for different countries on different dimensions using a keyword-based index generation process followed by normalization of the indexes (as seen in Figure 1). The keyword-based index generation process identifies social media posts related to a particular dimension of cyber by adding relevant keywords-based searches. For example, to identify cyber-related messages for Saudi Arabia, it would be required to search the captured cyber-related social media messages with keywords such as "KSA", or "Saudi". This would segregate the cyber-related messages that focus only on Saudi. For capturing cyber-attack-related posts, keywords such as "attack", "threat", "crisis", "problem", and "warfare" could be used on the already filtered cyber-related messages. By calculating the frequencies of such targeted messages against units of time (e.g., day, month, quarter, year, etc.), time-series indexes are generated for different countries on the cyber threat. This index-generation process was demonstrated previously by analyzing the multi-dimensional impact of COVID-19 [11].

For all these indexes, the minimum and maximum values could be significantly different since the numbers of social media posts varies greatly from one country to another. Hence, for the effective application of deep learning algorithms at subsequent stages, these cyber-threat indexes are required to be normalized with the following:

$$x_i = \frac{m_i - m_{min}}{m_{max} - m_{min}} \tag{1}$$

where $m_i$ represents the $i$th social media message on a cyber-related concern. Thus, Equation (1) generates all the country-wide indexes under same scale. Once multiple indexes are generated and normalized, deep learning methods such as anomaly detection using a convolutional neural network (CNN) can be applied.

### 2.5.3. Deep Learning

Anomaly detection algorithms fall within the deep learning techniques and have recently been used in [11,58–61,67–70] for detecting and identifying anomalies on rime-series data. The anomaly detector complements line charts by automatically detecting anomalies within time-series data. Using natural language processing (NLP)-based root-cause analysis, the anomaly detection process also explains the detected anomalies dynamically [71]. We have employed the anomaly detection algorithm to detect anomalies or abnormal cases of landslides in [67,68]. Apart from detecting abnormalities from the landslide data, research in [67,68] explained the root-causes of these detected anomalies in English using NLP-based technologies. The CNN-based anomaly detection process was also used in

analyzing tornadoes, as demonstrated in [69,70]. Moreover, as depicted in [58,59], disaster events can also be dynamically analyzed using anomaly detection on live social media feeds. Furthermore, the global news was critically analyzed using anomaly detection algorithms, as shown in [60,61]. More recently, the CNN-based anomaly detection algorithm was used to critically analyze the COVID-19 situation [11]. Within this section, we will look into the problem definition first, and then delve into the depth of anomaly detection process.

When a sequence of real values is presented, $x = x_1,\ x_2,\ x_3,\ \ldots,\ x_n,$ time-series anomaly detection's target produces an output sequence of $y = y_1,\ y_2,\ y,\ \ldots,\ y_n$, where $y_i \in \{0, 1\}$ denotes whether $x_i$ is an anomaly point.

Research in [72] demonstrated the process of saliency reduction (SR) from the visual saliency detection domain followed by the application of CNN to the output of the SR model. This study implements similar processes as described in [72] with the following three core tasks:

- Apply the Fourier transform for generating the log amplitude spectrum;
- Compute the SR;
- Apply the inverse Fourier transform to transform the sequence back to the spatial domain.

$$A(f) = Amplitude(f(x)) \tag{2}$$

$$P(f) = Phrase(f(x)) \tag{3}$$

$$L(f) = log(A(f)) \tag{4}$$

$$AL(f) = h_q(f){\cdot}L(f) \tag{5}$$

$$R(f) = L(f) - AL(f) \tag{6}$$

$$S(x) = \left\lVert f^{-1}(\exp(R(f) + iP(f))) \right\rVert \tag{7}$$

The Fourier transform and the inverse Fourier transform are represented by *f* and *f1*, respectively. Moreover, *x* represents the input sequence with shape nX1, and *A(f)* represents amplitude spectrum of sequence *x*. Furthermore, the phase spectrum of sequence *x* is denoted by *P(f)*. The log representation of *A(f)* is represented with *L(f)*; Then, the average spectrum of *L(f)* is presented with *AL(f)*, which can be estimated by convoluting the input sequence by $h_q(f)$. Then, $h_q(f)$ can be presented with a $q \times q$ matrix, as shown in Equation (8).

$$h_q(f) = \frac{1}{q^2}\begin{bmatrix} 1 & 1 & \ldots & 1 \\ 1 & 1 & \ldots & 1 \\ \ldots & \vdots & \ddots & 1 \\ 1 & 1 & \ldots & 1 \end{bmatrix} \tag{8}$$

As shown in Equation (6), *R(f)* is calculated by subtracting the averaged log spectrum *AL(f)* from the log spectrum *L(f)*. SR is denoted with *R(f)*. Finally, as shown in Equation (7), by applying an inverse Fourier transform, the sequence was assigned back to the spatial domain. The final output sequence, *S(x)*, represented within Equation (7), is called the saliency map [73]. The anomaly points are computed with Equation (9).

$$x = (\overline{x} + mean)(1 + var){\cdot}r + x \tag{9}$$

Within Equation (9), the local average of the preceding points is represented by $\overline{x}$. On the other hand, within Equation (9), mean and var are the mean and variance of all points within the current sliding window (i.e., randomly sampled r~N (0,1)). In this manner, CNN is employed to the saliency map (i.e., not on the raw input). Hence, the end-to-end process of anomaly detection harnesses efficiency and effectiveness [72,73]. As mentioned before, the anomaly detection algorithm implemented within this study uses NLP [71] to explain the root causes of all the anomalies in English.

### 2.5.4. Real-Time Visualization

Real-time visualization supports evidence-based, instant decision support to strategic users. The strategic users are required to make a wide range of informed decisions and decide on the national cyber posture. Since a strategic user could be mobile and away from desktop computers, they need to know about the cyber threat index on a wide range of devices such as mobile phones, tablets, and laptops. Hence, recent research in decision support systems support deployed iOS and Android apps, as shown in [11,58–62,67–70]. These interactive solutions allow strategic users to dynamically interact with the solution and obtain DL-driven insights on the available information. The methods proposed in this paper provide the first DL-based decision-making capability on countrywide cyber threats within mobile, tablet, and desktops (i.e., supporting iOS, Android, and Windows apps). Real-time visualizations were not depicted in [74–78].

This section describes how challenges from existing cyber intelligence systems were evaluated (i.e., Table 2), how requirements were developed (i.e., Table 3), and how solutions were built (i.e., Table 4) following the architecture development method (ADM) [24].

## 3. Results and Discussion

The presented method was tested and evaluated from 14 October 2022 to 27 December 2022 with a live Twitter feed. During that time, about 15,983 tweets from 15,315 users in 47 different languages were identified by our data gathering process. A range of AI-based analyses, such as language detection, translation, and sentiment analyses were performed on these tweets. In addition, 3718 translations were performed on non-English messages followed by sentiment analysis. For testing each step of the presented method, different technological components were used, as depicted in Table 5. As seen from Table 5, the "data gathering and preprocessing" stage mainly used Microsoft Power Automate [64] along with Azure Cognitive Services [66] The "data manipulation stage" predominantly used Microsoft SQL Server and Microsoft Power BI desktop-based technologies [11]. Then, the "deep learning" stage was implemented with a CNN-based anomaly detection algorithm along with an NLP [71]-based root cause analysis. The deep learning stage was validated using the Microsoft Power BI Desktop. Finally, the real-time visualization aspect of the presented methods was deployed and validated using both the Microsoft Power BI Desktop and Microsoft Power BI Services. As seen from Table 2, the visualization stage covered the Windows-based dashboard, the Web interface, the iOS app, as well as the Android app on different form factors.

Table 6 demonstrates the method validity for the "data gathering and prepressing" stage (of Figure 1). As seen in Table 6, the daily average of sentiment reveals that social media users were most negative (i.e., concerned and critical) about cyber-related issues on 8 November 2022, with an average negative sentiment confidence of 0.477. Using a named entity detection (NER) process, the reason for this social media outcry on the cyber issue could be drilled down into. Moreover, questions such as where cyber concerns are generated could also be found from the identified location. However, Table 6 reports this location at an aggregated manner, maintaining the privacy of the social media users. Table 7 demonstrates the method validity for the "data manipulation" stage of Figure 2. As seen in Table 7, cyber indexes for six different countries (i.e., China, Australia, Russia, Ukraine, Iran, and India) for 75 days were generated (from 14 October 2022 to 27 December 2022). For generating these indexes, data gathered and prepared at the earlier stage was utilized. Fluctuations of these index values provide an understanding of cyber-related concerns for a particular country on a specific day. Higher index values correspond to a higher level of cyber-related concerns for the selected country. As seen in Table 8, with the monitored period (i.e., from 14 October 2022 to 27 December 2022), cyber-related concerns for Russia was at its highest on 30 October 2022 (with the value of the cyber index for Russia being 13 on 30 October 2022). This particular index was also highlighted as an anomaly by the subsequent deep learning process, as depicted in Figure 3. Using this information, strategic decision makers can adjust their national cyber posture to mitigate

the cyber threat on the specific days that were automatically highlighted as anomalies. Using the same process, time-series indexes could be generated on any unit of time (e.g., daily, monthly, quarterly, or yearly). As seen from Table 5, the index-generation process could use either a SQL Server (with SQL statements) or Microsoft Power BI (with DAX expression). Code snippets displayed within Code 1 and Code 2 demonstrates the keyword-based index-generation process in SQL Statements as well as in DAX expressions. The "data manipulation" subsection of the "method details" section highlights that, apart from creating countrywide cyber indexes, cyber indexes on any other concerns could be created by the methodologies described within this paper. For example, to create a global index on cyber-attack related concerns, keywords such as "attack", "threat", "crisis", "problem", and "warfare" could be used. Code 3 demonstrates the necessary SQL statement to create such an index. Thus, individual indexes representing different concerns could be generated (e.g., Code 3).

**Table 5.** Technology components used for validating the cyber threat index generation method.

| Features | Microsoft Power Automate | Azure Cognitive Services | Microsoft SQL Server | Microsoft Power BI Desktop | Microsoft Power BI Services |
|---|:---:|:---:|:---:|:---:|:---:|
| Obtain cyber social media posts | • | | | | |
| Detect language | • | • | | | |
| Translate to English | • | • | | | |
| Sentiment analysis | • | • | | | |
| Generate cyber threat index | | | • | • | |
| Normalize index | | | • | • | |
| Detect anomalies (CNN) | | | | • | |
| Explain anomalies (NLP) | | | | • | |
| Dashboard for Windows | | | | • | • |
| Web Access on mobile, tablet, or desktop | | | | | • |
| iOS app on mobile/tablet | | | | | • |
| Android app on mobile/tablet | | | | | • |

**Table 6.** AI-based aggregation and analysis of cyber-related tweets.

| Date | Tweet IDs | User IDs | Locations | Languages | Retweets | Avg. -ve Sentiment | Avg. Neutral Sentiment | Avg. +ve Sentiment | Translations |
|---|---|---|---|---|---|---|---|---|---|
| 10/14/2022 | 211 | 189 | 122 | 15 | 77,089 | 0.399 | 0.402 | 0.199 | 67 |
| 10/15/2022 | 219 | 208 | 116 | 18 | 408,635 | 0.292 | 0.461 | 0.247 | 74 |
| 10/16/2022 | 208 | 205 | 111 | 18 | 428,407 | 0.312 | 0.442 | 0.246 | 67 |
| 10/17/2022 | 221 | 208 | 122 | 14 | 188,791 | 0.295 | 0.464 | 0.240 | 60 |
| 10/18/2022 | 186 | 180 | 101 | 18 | 49,255 | 0.315 | 0.492 | 0.193 | 56 |
| 10/19/2022 | 226 | 219 | 133 | 18 | 132,222 | 0.354 | 0.421 | 0.225 | 55 |
| 10/20/2022 | 216 | 215 | 123 | 17 | 231,915 | 0.321 | 0.469 | 0.210 | 51 |
| 10/21/2022 | 206 | 204 | 129 | 17 | 533,082 | 0.434 | 0.413 | 0.152 | 37 |
| 10/22/2022 | 219 | 209 | 118 | 14 | 134,067 | 0.409 | 0.401 | 0.190 | 46 |
| 10/23/2022 | 223 | 207 | 116 | 18 | 342,49 | 0.333 | 0.466 | 0.200 | 69 |

**Table 6.** *Cont.*

| Date | Tweet IDs | User IDs | Locations | Languages | Retweets | Avg. -ve Sentiment | Avg. Neutral Sentiment | Avg. +ve Sentiment | Translations |
|---|---|---|---|---|---|---|---|---|---|
| 10/24/2022 | 226 | 218 | 128 | 16 | 88,944 | 0.440 | 0.353 | 0.206 | 59 |
| 10/25/2022 | 227 | 219 | 118 | 20 | 200,700 | 0.434 | 0.403 | 0.164 | 46 |
| 10/26/2022 | 219 | 205 | 113 | 13 | 30,097 | 0.375 | 0.413 | 0.211 | 48 |
| 10/27/2022 | 222 | 219 | 121 | 14 | 175,143 | 0.339 | 0.423 | 0.238 | 47 |
| 10/28/2022 | 218 | 212 | 124 | 14 | 287,112 | 0.388 | 0.377 | 0.235 | 48 |
| 10/29/2022 | 224 | 215 | 126 | 14 | 176,450 | 0.414 | 0.356 | 0.230 | 41 |
| 10/30/2022 | 222 | 215 | 114 | 12 | 217,949 | 0.345 | 0.452 | 0.202 | 48 |
| 10/31/2022 | 209 | 205 | 113 | 18 | 252,942 | 0.322 | 0.480 | 0.199 | 55 |
| 11/1/2022 | 227 | 223 | 133 | 14 | 175,690 | 0.312 | 0.440 | 0.248 | 48 |
| 11/2/2022 | 225 | 216 | 120 | 19 | 158,510 | 0.370 | 0.445 | 0.184 | 54 |
| 11/3/2022 | 219 | 213 | 126 | 15 | 435,121 | 0.464 | 0.384 | 0.150 | 47 |
| 11/4/2022 | 227 | 214 | 114 | 17 | 178,945 | 0.345 | 0.413 | 0.242 | 48 |
| 11/5/2022 | 219 | 208 | 123 | 12 | 65,565 | 0.446 | 0.360 | 0.194 | 53 |
| 11/6/2022 | 212 | 205 | 105 | 16 | 469,544 | 0.364 | 0.408 | 0.228 | 47 |
| 11/7/2022 | 221 | 210 | 107 | 13 | 89,628 | 0.379 | 0.419 | 0.203 | 43 |
| 11/8/2022 | 226 | 221 | 117 | 14 | 115,866 | 0.477 | 0.352 | 0.171 | 47 |
| 11/9/2022 | 213 | 205 | 117 | 19 | 73,431 | 0.430 | 0.376 | 0.193 | 49 |
| 11/10/2022 | 212 | 207 | 124 | 15 | 90,221 | 0.363 | 0.410 | 0.226 | 42 |
| 11/11/2022 | 216 | 213 | 109 | 12 | 110,456 | 0.334 | 0.432 | 0.233 | 40 |
| 11/12/2022 | 217 | 213 | 121 | 14 | 104,071 | 0.465 | 0.364 | 0.170 | 41 |
| 11/13/2022 | 210 | 201 | 109 | 17 | 89,242 | 0.439 | 0.378 | 0.182 | 51 |
| 11/14/2022 | 216 | 210 | 136 | 14 | 230,263 | 0.260 | 0.402 | 0.338 | 36 |
| 11/15/2022 | 225 | 214 | 132 | 14 | 255,924 | 0.349 | 0.377 | 0.274 | 48 |
| 11/16/2022 | 215 | 207 | 121 | 14 | 103,983 | 0.378 | 0.386 | 0.235 | 52 |
| 11/17/2022 | 218 | 212 | 98 | 13 | 226,855 | 0.263 | 0.450 | 0.286 | 52 |
| 11/18/2022 | 221 | 215 | 91 | 12 | 692,947 | 0.328 | 0.388 | 0.283 | 44 |
| 11/19/2022 | 220 | 215 | 103 | 11 | 267,758 | 0.250 | 0.542 | 0.207 | 37 |
| 11/20/2022 | 215 | 203 | 98 | 18 | 398,197 | 0.272 | 0.546 | 0.180 | 54 |
| 11/21/2022 | 223 | 215 | 112 | 18 | 793,534 | 0.309 | 0.408 | 0.281 | 46 |
| 11/22/2022 | 205 | 199 | 104 | 12 | 276,979 | 0.291 | 0.354 | 0.355 | 36 |
| 11/23/2022 | 212 | 209 | 102 | 11 | 229,434 | 0.334 | 0.469 | 0.196 | 34 |
| 11/24/2022 | 218 | 209 | 110 | 13 | 571,039 | 0.325 | 0.459 | 0.215 | 54 |
| 11/25/2022 | 209 | 202 | 108 | 17 | 26,714 | 0.339 | 0.414 | 0.246 | 48 |
| 11/26/2022 | 214 | 208 | 120 | 14 | 95,425 | 0.361 | 0.378 | 0.261 | 47 |
| 11/27/2022 | 199 | 196 | 110 | 15 | 67,676 | 0.326 | 0.427 | 0.247 | 53 |
| 11/28/2022 | 200 | 196 | 99 | 14 | 53,436 | 0.254 | 0.493 | 0.254 | 49 |
| 11/29/2022 | 213 | 212 | 131 | 14 | 1,868,246 | 0.206 | 0.596 | 0.198 | 36 |
| 11/30/2022 | 203 | 197 | 116 | 15 | 1,667,156 | 0.250 | 0.508 | 0.240 | 36 |
| 12/1/2022 | 211 | 195 | 113 | 17 | 695,611 | 0.372 | 0.372 | 0.254 | 67 |
| 12/2/2022 | 198 | 189 | 102 | 21 | 142,491 | 0.353 | 0.432 | 0.215 | 52 |
| 12/3/2022 | 207 | 203 | 112 | 16 | 182,689 | 0.456 | 0.372 | 0.172 | 37 |
| 12/4/2022 | 190 | 175 | 95 | 14 | 101,186 | 0.415 | 0.404 | 0.181 | 42 |
| 12/5/2022 | 189 | 187 | 112 | 13 | 561,479 | 0.347 | 0.474 | 0.178 | 42 |
| 12/6/2022 | 198 | 192 | 88 | 14 | 77,006 | 0.339 | 0.393 | 0.267 | 38 |

**Table 6.** *Cont.*

| Date | Tweet IDs | User IDs | Locations | Languages | Retweets | Avg. -ve Sentiment | Avg. Neutral Sentiment | Avg. +ve Sentiment | Translations |
|---|---|---|---|---|---|---|---|---|---|
| 12/7/2022 | 201 | 174 | 96 | 14 | 20,646 | 0.349 | 0.434 | 0.216 | 38 |
| 12/8/2022 | 210 | 199 | 108 | 18 | 26,394 | 0.411 | 0.401 | 0.187 | 56 |
| 12/9/2022 | 207 | 202 | 110 | 16 | 40,478 | 0.380 | 0.406 | 0.213 | 47 |
| 12/10/2022 | 197 | 194 | 93 | 18 | 104,133 | 0.426 | 0.355 | 0.219 | 58 |
| 12/11/2022 | 211 | 197 | 99 | 15 | 61,179 | 0.451 | 0.332 | 0.217 | 60 |
| 12/12/2022 | 210 | 206 | 112 | 18 | 70,526 | 0.348 | 0.454 | 0.198 | 53 |
| 12/13/2022 | 213 | 209 | 136 | 16 | 73,302 | 0.367 | 0.454 | 0.179 | 57 |
| 12/14/2022 | 204 | 189 | 99 | 16 | 36,787 | 0.337 | 0.436 | 0.228 | 56 |
| 12/15/2022 | 218 | 192 | 116 | 14 | 35,915 | 0.291 | 0.449 | 0.259 | 58 |
| 12/16/2022 | 197 | 177 | 97 | 12 | 449,725 | 0.456 | 0.350 | 0.194 | 45 |
| 12/17/2022 | 213 | 205 | 117 | 18 | 153,354 | 0.433 | 0.374 | 0.193 | 51 |
| 12/18/2022 | 215 | 199 | 114 | 16 | 91,958 | 0.385 | 0.396 | 0.218 | 52 |
| 12/19/2022 | 207 | 195 | 105 | 12 | 974,928 | 0.330 | 0.435 | 0.234 | 36 |
| 12/20/2022 | 211 | 204 | 94 | 14 | 531,816 | 0.330 | 0.482 | 0.188 | 39 |
| 12/21/2022 | 209 | 205 | 114 | 16 | 158,890 | 0.339 | 0.517 | 0.144 | 43 |
| 12/22/2022 | 192 | 187 | 87 | 13 | 224,755 | 0.411 | 0.405 | 0.183 | 40 |
| 12/23/2022 | 214 | 201 | 107 | 14 | 517,649 | 0.324 | 0.409 | 0.266 | 57 |
| 12/24/2022 | 205 | 194 | 101 | 17 | 56,815 | 0.313 | 0.419 | 0.267 | 60 |
| 12/25/2022 | 227 | 210 | 118 | 18 | 35,784 | 0.254 | 0.411 | 0.335 | 60 |
| 12/26/2022 | 226 | 208 | 108 | 16 | 925,420 | 0.260 | 0.401 | 0.339 | 65 |
| 12/27/2022 | 231 | 207 | 107 | 11 | 98,908 | 0.320 | 0.436 | 0.243 | 63 |

**Table 7.** Cyber index generated for six different countries.

| Tweet Date | Cyber Index China | Cyber Index Australia | Cyber Index Russia | Cyber Index Ukraine | Cyber Index Iran | Cyber Index India |
|---|---|---|---|---|---|---|
| 10/14/2022 | 0 | 0 | 0 | 0 | 1 | 3 |
| 10/15/2022 | 0 | 0 | 4 | 2 | 1 | 4 |
| 10/16/2022 | 1 | 2 | 4 | 3 | 1 | 1 |
| 10/17/2022 | 0 | 0 | 2 | 1 | 2 | 4 |
| 10/18/2022 | 1 | 0 | 8 | 1 | 2 | 1 |
| 10/19/2022 | 0 | 1 | 3 | 0 | 5 | 3 |
| 10/20/2022 | 1 | 2 | 3 | 1 | 4 | 1 |
| 10/21/2022 | 0 | 4 | 4 | 1 | 1 | 1 |
| 10/22/2022 | 0 | 1 | 1 | 0 | 0 | 2 |
| 10/23/2022 | 1 | 0 | 2 | 3 | 3 | 3 |
| 10/24/2022 | 2 | 0 | 5 | 0 | 4 | 2 |
| 10/25/2022 | 1 | 2 | 1 | 0 | 1 | 0 |
| 10/26/2022 | 0 | 1 | 0 | 1 | 1 | 1 |
| 10/27/2022 | 2 | 0 | 0 | 0 | 3 | 4 |
| 10/28/2022 | 0 | 1 | 1 | 0 | 1 | 4 |
| 10/29/2022 | 2 | 2 | 2 | 1 | 1 | 3 |
| 10/30/2022 | 6 | 2 | 13 | 1 | 0 | 2 |
| 10/31/2022 | 1 | 0 | 4 | 0 | 0 | 0 |
| 11/1/2022 | 1 | 0 | 6 | 3 | 1 | 3 |
| 11/2/2022 | 1 | 0 | 2 | 3 | 8 | 1 |

**Table 7.** *Cont.*

| Tweet Date | Cyber Index China | Cyber Index Australia | Cyber Index Russia | Cyber Index Ukraine | Cyber Index Iran | Cyber Index India |
|---|---|---|---|---|---|---|
| 11/3/2022 | 0 | 1 | 0 | 3 | 1 | 2 |
| 11/4/2022 | 0 | 1 | 1 | 1 | 0 | 5 |
| 11/5/2022 | 0 | 0 | 1 | 2 | 0 | 2 |
| 11/6/2022 | 1 | 1 | 2 | 2 | 1 | 2 |
| 11/7/2022 | 1 | 2 | 3 | 1 | 2 | 5 |
| 11/8/2022 | 1 | 1 | 1 | 0 | 3 | 3 |
| 11/9/2022 | 1 | 0 | 3 | 1 | 0 | 1 |
| 11/10/2022 | 1 | 2 | 2 | 0 | 1 | 0 |
| 11/11/2022 | 2 | 2 | 8 | 3 | 1 | 2 |
| 11/12/2022 | 1 | 2 | 2 | 0 | 1 | 5 |
| 11/13/2022 | 1 | 1 | 1 | 3 | 1 | 2 |
| 11/14/2022 | 1 | 0 | 0 | 2 | 0 | 6 |
| 11/15/2022 | 1 | 1 | 5 | 1 | 0 | 1 |
| 11/16/2022 | 0 | 3 | 8 | 2 | 2 | 2 |
| 11/17/2022 | 0 | 0 | 0 | 0 | 2 | 1 |
| 11/18/2022 | 0 | 0 | 1 | 1 | 1 | 4 |
| 11/19/2022 | 0 | 0 | 3 | 4 | 0 | 2 |
| 11/20/2022 | 1 | 2 | 2 | 1 | 2 | 2 |
| 11/21/2022 | 0 | 1 | 1 | 0 | 2 | 0 |
| 11/22/2022 | 0 | 0 | 0 | 0 | 1 | 2 |
| 11/23/2022 | 1 | 0 | 4 | 0 | 2 | 1 |
| 11/24/2022 | 1 | 0 | 3 | 1 | 2 | 1 |
| 11/25/2022 | 0 | 0 | 1 | 0 | 0 | 2 |
| 11/26/2022 | 0 | 0 | 2 | 3 | 0 | 0 |
| 11/27/2022 | 0 | 0 | 0 | 0 | 3 | 2 |
| 11/28/2022 | 0 | 0 | 1 | 1 | 0 | 0 |
| 11/29/2022 | 2 | 0 | 1 | 2 | 1 | 1 |
| 11/30/2022 | 1 | 0 | 0 | 1 | 0 | 1 |
| 12/1/2022 | 1 | 0 | 2 | 1 | 1 | 1 |
| 12/2/2022 | 1 | 1 | 1 | 2 | 1 | 4 |
| 12/3/2022 | 1 | 0 | 4 | 0 | 0 | 2 |
| 12/4/2022 | 1 | 1 | 6 | 3 | 1 | 1 |
| 12/5/2022 | 0 | 3 | 4 | 1 | 2 | 1 |
| 12/6/2022 | 0 | 0 | 1 | 1 | 3 | 1 |
| 12/7/2022 | 2 | 0 | 2 | 2 | 1 | 2 |
| 12/8/2022 | 4 | 3 | 4 | 0 | 2 | 2 |
| 12/9/2022 | 2 | 4 | 1 | 0 | 2 | 2 |
| 12/10/2022 | 0 | 0 | 1 | 1 | 2 | 1 |

**Table 7.** *Cont.*

| Tweet Date | Cyber Index China | Cyber Index Australia | Cyber Index Russia | Cyber Index Ukraine | Cyber Index Iran | Cyber Index India |
|---|---|---|---|---|---|---|
| 12/11/2022 | 0 | 0 | 3 | 2 | 2 | 0 |
| 12/12/2022 | 3 | 1 | 2 | 1 | 1 | 2 |
| 12/13/2022 | 0 | 0 | 4 | 2 | 0 | 4 |
| 12/14/2022 | 2 | 0 | 2 | 0 | 1 | 1 |
| 12/15/2022 | 3 | 1 | 0 | 0 | 0 | 4 |
| 12/16/2022 | 2 | 0 | 1 | 0 | 0 | 2 |
| 12/17/2022 | 2 | 0 | 1 | 0 | 0 | 0 |
| 12/18/2022 | 1 | 0 | 1 | 3 | 4 | 1 |
| 12/19/2022 | 2 | 0 | 6 | 1 | 0 | 1 |
| 12/20/2022 | 0 | 0 | 10 | 1 | 2 | 0 |
| 12/21/2022 | 1 | 0 | 9 | 1 | 0 | 2 |
| 12/22/2022 | 0 | 0 | 2 | 2 | 1 | 0 |
| 12/23/2022 | 3 | 0 | 1 | 1 | 1 | 0 |
| 12/24/2022 | 1 | 0 | 1 | 1 | 0 | 3 |
| 12/25/2022 | 1 | 1 | 2 | 2 | 2 | 1 |
| 12/26/2022 | 1 | 1 | 1 | 0 | 0 | 2 |
| 12/27/2022 | 2 | 0 | 1 | 2 | 1 | 1 |

**Table 8.** Numbers of anomalies detected with different percentages of sensitivity.

| Sensitivity of Anomaly Detection | China | Australia | Russia | Ukraine | Iran | India |
|---|---|---|---|---|---|---|
| 70% | 2 | 4 | 3 | 2 | 3 | 1 |
| 80% | 3 | 5 | 4 | 3 | 3 | 1 |
| 90% | 3 | 6 | 4 | 3 | 3 | 1 |
| 100% | 3 | 6 | 5 | 3 | 3 | 1 |

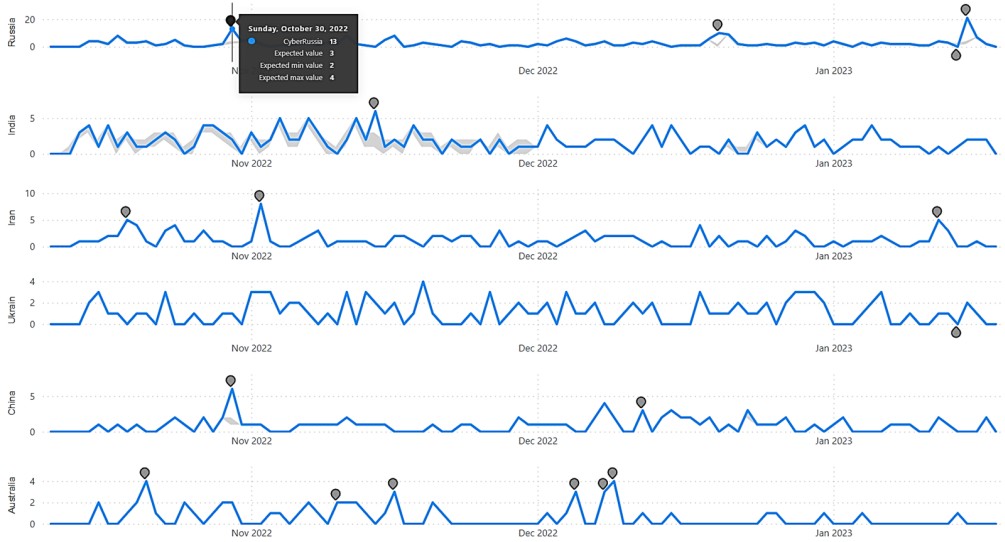

**Figure 3.** CNN-based anomaly detection executed on cyber threat indexes within a Windows environment.

Once the cyber threat indexes are generated, the deep learning stage performs CNN-based anomaly detection, as shown in Figure 3. As shown in Figure 3, anomaly detection highlights a number of anomalies for all index values for the six selected countries shown in Table 7. China, Australia, Russia, Ukraine, Iran, and India had 2, 4, 3, 2, 3, and 1 anomalies detected by the anomaly detection algorithm with 70% sensitivity. By adjusting to a higher level of sensitivity, a higher number of anomalies could be observed by the anomaly detection algorithm, as seen in Table 8. For example, even though four anomalies were detected for Australia with 70% sensitivity (as shown in Figure 2), six anomalies were detected with 100% sensitivity (shown in Table 8). It should be noted that changes in sensitivity did not have any impact on the cyber threat indexes for Iran and India. Both Iran and India retained the same number of anomalies through sensitivity parameterization of 70%, 80%, 90%, and 100%. Figure 3 demonstrates deployment on the Windows desktop. The real-time visualizations were also deployed as an Android app (i.e., Figure 4) and an iOS app (i.e., Figure 5). As seen from both Figures 4 and 5, the deployed app provides interactive decision-making capabilities, as the strategic user can select any time duration and the cyber threat concerns are directly updated in the dashboard according to the selected timeframe. While Figure 4 shows deployment through the Power BI Mobile App in the Android environment, Figure 5 depicts deployment within the iOS environment. Since both Figures 4 and 5 show deployment on different mobile platforms, the contents might be difficult for viewing. Hence, a publicly accessible web link is available at https://app.powerbi.com/view?r=eyJrIjoiYWJjOGY5YTUtZDBlNy00MTg1LWFkMTM tM2RmYzYzODQ1NzE1IiwidCI6IjBkMWI4YmRlLWZmYzEtNGY1Yy05NjAwLTJhNzUzZ GFjYmEwNSJ9&pageName=ReportSection, accessed on 19 February 2023.

---

**Code 1: Cyber Index for Russia with SQL Statement**

*SELECT * FROM [dbo].Tweets WHERE TweetSourceType='CYBER' AND (TweetText LIKE '%RUSSIA%' OR TranslatedText LIKE '%RUSSIA%' ) ORDER BY Time*

---

**Code 2: Cyber Index for Russia with DAX Expression**

*CyberRussia = IF(COUNTROWS(FILTER(CyberSocial, OR(CONTAINSSTRING(CyberSocial[TweetText], "Russia"), CONTAINSSTRING(CyberSocial[TranslatedText], "Russia"))))=BLANK(), 0, COUNTROWS(FILTER(CyberSocial, OR(CONTAINSSTRING(CyberSocial[TweetText], "Russia"), CONTAINSSTRING(CyberSocial[TranslatedText], "Russia")))))*

---

**Code 3: Index Focusing on Cyber-Related Threats/Crisis/Issue SQL Statement**

*SELECT * FROM [dbo].Tweets WHERE TweetSourceType='CYBER' AND ((TweetText LIKE '%crisis%' or TweetText LIKE '%threat%' or TweetText LIKE '%issue%' or TweetText LIKE '%problem%' or TweetText LIKE '%crime%' or TweetText LIKE '%attack%' or TweetText LIKE '%warfare%' or TweetText LIKE '%alert%'or TweetText LIKE '%warn%') OR (TranslatedText LIKE '%crisis%' or TranslatedText LIKE '%threat%' or TranslatedText LIKE '%issue%' or TranslatedText LIKE '%problem%' or TranslatedText LIKE '%crime%' or TranslatedText LIKE '%attack%' or TranslatedText LIKE '%warfare%' or TranslatedText LIKE '%alert%'or TweetText LIKE '%warn%')) ORDER BY Time*

---

It should be reiterated that CNN was implemented with low-code solutions provided by the Microsoft Power Platform [65]. This low-code application of CNN allows seamless mobile deployment of CNN (as shown in Figures 4 and 5) without writing any device-specific codes (i.e., targeting iOS or Android). As a result, CNN has been popularly used to solve deep learning problems in the areas of landslides, tornadoes, global events, and social media analysis, as shown in [11,58–61,67–70].

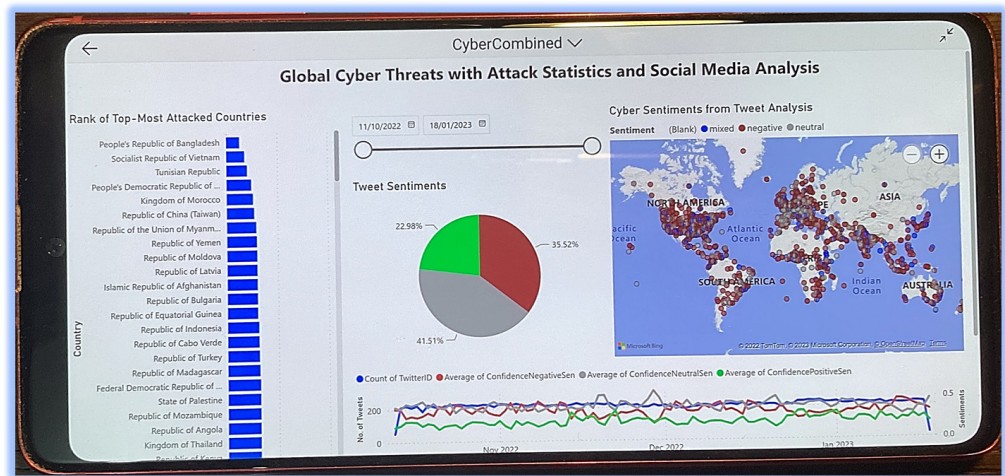

**Figure 4.** Social media-based cyber threat dashboard executed on a Samsung Galaxy Note 10 Lite Mobile (Android Device).

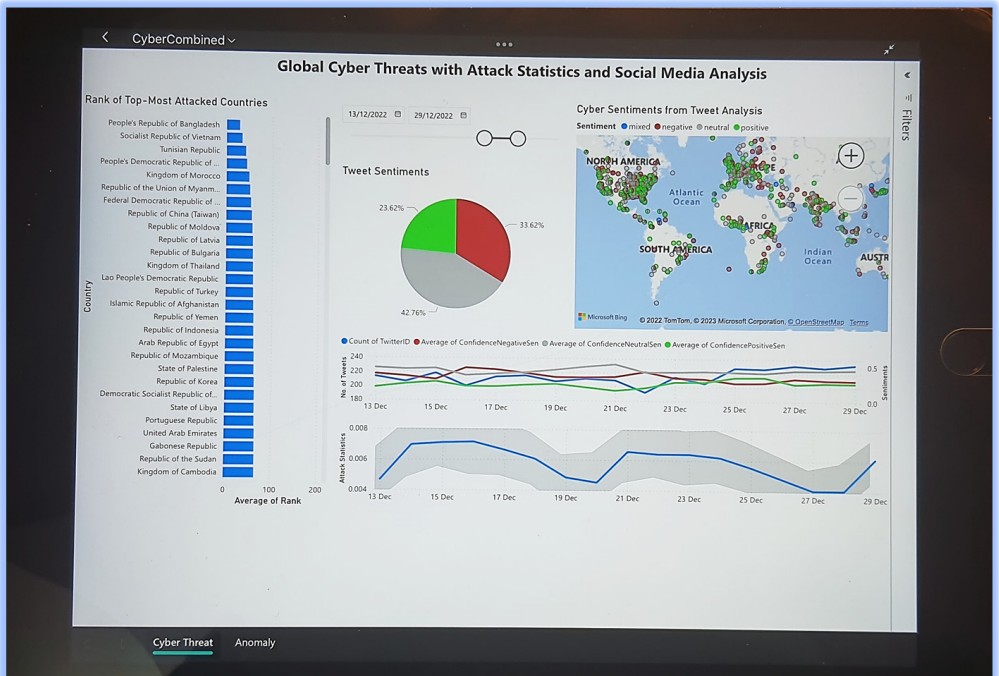

**Figure 5.** Social media-based cyber threat dashboard executed on an Apple iPad 9th Generation (iOS Device).

## 4. Conclusions

This paper proposed a methodological approach of applying AI-based algorithms with CNN-based anomaly detection to perform the following:

- Autonomously identify, analyze, and report a new cyber threat index for any countries;
- Autonomously identify, analyze, and report global cyber threat anomalies;
- The presented methodology was validated with live Twitter feeds for 75 consecutive days starting from 14 October 2022 to 27 December 2022;
- AI-based language detection, translation, and sentiment analysis on 15,983 tweets in 47 different languages (unlike [14–16], which only worked in a single language);
- 75 days of daily cyber threat indexes generated for China, Australia, Russia, Ukraine, Iran, and India;

- CNN-based anomaly detection automatically detecting 2, 4, 3, 2, 3, and 1 anomaly for China, Australia, Russia, Ukraine, Iran, and India, respectively;
- CNN-based anomaly detection validated under hyper parameterization of sensitivity percentages at 70%, 80%, 90%, and 100%;
- Interactive cyber threat analysis solution deployed on desktop, mobile, and tablet environments with Windows, iOS, and Android apps (unlike [14–16], which only worked on desktops and laptops).

The concept of a countrywide cyber threat index is innovative and would allow instant understanding of the overall level of cyber threat being experienced by a country on any given timeframe (i.e., day, month, year, etc.). The methodology presented within this study would allow strategic decision makers to adjust national-level cyber posture with informed and evidence-based decisions.

However, since the methodology described within this study explicitly relies on social media data for creating the index (and normalizing the indexes with Equation (1)), any threats to social media platforms or infrastructure would be a threat to the presented system. For example, if the Twitter platform is under cyber-attack, this presented system would fail. In another example, if a group of people or organization conducts an information operation by generating a huge number of fake posts (i.e., related to cyber), then that would also be a threat to the presented system.

In the future, we would endeavor to create more cyber-related indexes focusing on cyber offence, cyber defence, and other cyber dimensions. Moreover, we would apply a wide variety of DL methods, such as the recurrent neural network (RNN), generative adversarial network (GAN), radial basis function network (RBFN), and others, on multiple dimensions of cyber indexes.

**Funding:** This research received no external funding.

**Data Availability Statement:** All the source files (including the CyberCombined.pbix Microsoft Power BI solution, *.csv files corresponding to AI-based Tweet analysis, countrywide cyber threat indexes, and the SQL_Statements.txt file for SQL statements) are located at https://github.com /DrSufi/CyberThreatIndex, accessed on 19 February 2023). The Microsoft Power BI solution file (CyberCombined.pbix) can be opened after installing the Microsoft Power BI Desktop (freely available at https://www.microsoft.com/en-us/download/details.aspx?id=58494, accessed on 19 February 2023).

**Acknowledgments:** The author would like to thank Edris Alam of the Rabdan Academy for evaluating and providing his valuable feedback on the deployed app.

**Conflicts of Interest:** The author declares no conflict of interest.

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
