# Peer review of "A New Social Media-Driven Cyber Threat Intelligence"

_electronics, doi:10.3390/electronics12051242_

Round 1

Reviewer 1 Report

This paper describes an innovative use of social media driven intelligence on defending cyber threats. The methodological approach completed a systematic literature review and by investigating the challenges faced by existing literature, the author proposes an AI, ML, and deep learning (i.e., with Convolutional Neural Network) based solution for creating a threat index. I believe that the paper contributes to existing body of knowledge in the following way:

1)      A new cyber threat index is proposed.

2)      The threat index was analyzed with CNN based deep learning to identify anomalies (i.e., incidents of cyber threat).

3)      The methodology uses modern AI and NLP based techniques like Sentiment analysis, language translation to comprehend multilingual social media posts on Cyber issues.

4)      This innovative methodology was tested with live twitter feed for 75 days to generate threat indexes for China, Australia, Russia, Ukraine, Iran, and India

5)      This techniques was deployed and evaluated in multiple platforms like windows, android, and iOS

However, the manuscript could be improved further with following:

1)      The systematic approach demonstrated in Fig. 1, Table 2, Table 3, Table 4 means literature review mapped to challenges, challenges mapped to a set of requirements, and requirements mapped to solution. The author should also discuss briefly about applicable Enterprise Architecture methodology or Architectural Development Method (ADM) that could be related to the systematic approach described within this paper. This would provide a software engineering perspective on this study.

2)      The author should clarify whether Fig. 4 and Fig. 5 are deployed mobile app or power bi deployment, or webpage viewed in mobile. Is there a live web link available to view this information?

3)      What are the scenarios that could be a threat to this proposed system? If there are threats to this approach then the author should discuss it briefly within the conclusion section. This would create another future research direction.

4)      Please discuss within discussion section, how other researchers could generate new indexes tailored to their own requirements

5)      How was CNN applied? 

Author Response

This paper describes an innovative use of social media driven intelligence on defending cyber threats. The methodological approach completed a systematic literature review and by investigating the challenges faced by existing literature, the author proposes an AI, ML, and deep learning (i.e., with Convolutional Neural Network) based solution for creating a threat index. I believe that the paper contributes to existing body of knowledge in the following way:

1)      A new cyber threat index is proposed.

2)      The threat index was analyzed with CNN based deep learning to identify anomalies (i.e., incidents of cyber threat).

3)      The methodology uses modern AI and NLP based techniques like Sentiment analysis, language translation to comprehend multilingual social media posts on Cyber issues.

4)      This innovative methodology was tested with live twitter feed for 75 days to generate threat indexes for China, Australia, Russia, Ukraine, Iran, and India

5)      This techniques was deployed and evaluated in multiple platforms like windows, android, and iOS

Response: Many thanks for considering this work as innovative and contributing to existing body of knowledge. I really appreciated your feedback.

However, the manuscript could be improved further with following:

1)      The systematic approach demonstrated in Fig. 1, Table 2, Table 3, Table 4 means literature review mapped to challenges, challenges mapped to a set of requirements, and requirements mapped to solution. The author should also discuss briefly about applicable Enterprise Architecture methodology or Architectural Development Method (ADM) that could be related to the systematic approach described within this paper. This would provide a software engineering perspective on this study.

Response: Many thanks for this suggestion. To address this issue, following statements have been added to the updated manuscript within Section 2:

“Thus, the system development framework stipulated by The Open Group Architectural Framework (TOGAF) and Archimate standard were broadly followed [24].”

“This section described how challenges from existing cyber intelligence systems were evaluated (i.e., Table 2), how requirements were developed (i.e., Table 3), and how solutions were built (i.e., Table 4) following Architecture Development Method (ADM) [24].”

2)      The author should clarify whether Fig. 4 and Fig. 5 are deployed mobile app or power bi deployment, or webpage viewed in mobile. Is there a live web link available to view this information?

Response: Many thanks for this suggestion. To address this issue, following statements have been added to the updated manuscript within Section 3:

“While Figure 4 shows deployment through Power BI Mobile App on Android environ-ment, Figure 5 depicts deployment within iOS environment. Since both Figure 4 and Fig-ure 5 shows deployment within different mobile platforms, the contents might be difficult for viewing. Hence, a publicly accessible web link is available at https://app.powerbi.com/view?r=eyJrIjoiYWJjOGY5YTUtZDBlNy00MTg1LWFkMTMtM2RmYzYzODQ1NzE1IiwidCI6IjBkMWI4YmRlLWZmYzEtNGY1Yy05NjAwLTJhNzUzZGFjYmEwNSJ9&pageName=ReportSection”.

3)      What are the scenarios that could be a threat to this proposed system? If there are threats to this approach, then the author should discuss it briefly within the conclusion section. This would create another future research direction.

Response: Many thanks for this suggestion. To address this issue, following statements have been added to the updated manuscript within Section 2:

“However, since the methodology described within this study explicitly relies on Social media data for creating the index (and normalizing the indexes with Eq. 1), any threats to social media platforms or infrastructure would be a threat to the presented sys-tem. For example, if Twitter platform is under cyber-attack, this presented system would fail. In another example, if a group of people or organization conduct information operation by generating a huge number of fake posts (i.e., related to cyber), then that would also be a threat to the presented system.”

4)      Please discuss within discussion section, how other researchers could generate new indexes tailored to their own requirements

Response: Many thanks for this suggestion. To address this issue, the caption of Code 3 was updated. Moreover, following statements have been added to the updated manuscript within Section 3: 

“The “Data Manipulation” subsection of the “Method Details” section highlighted that apart from creating country-wise cyber indexes, cyber indexes on any other concerns could be created by the methodologies described within this paper. For example, to create a global index on cyber-Attack related concerns, keywords like “attack”, “threat”, “crisis”, “problem”, “warfare” could be used. Code 3 demonstrates the necessary SQL statement to create such index. Thus, individual indexes representing different concerns could be generated (e.g., Code 3).”

5)      How was CNN applied? 

Response: Many thanks for this suggestion. To address this issue, following statements have been added to the updated manuscript within Section 3:

“It should be reiterated that CNN was implemented with low-code solutions provided by Microsoft Power Platform [65]. This low-code application of CNN allows seamless mobile deployment of CNN (as shown in Figure 4 and Figure 5) without writing any de-vice specific codes (i.e., targeting iOS or Android). As a result, CNN have been in popular-ly used in solving deep learning problems in the area of landslides, tornado, global events, and social media analysis as shown in [11] [58] [59] [60] [61] [67] [68] [69] [70].”

Reviewer 2 Report

ü  Re. the manuscript “A New Social Media Driven Electronic-Warfare-Threat Intelligence with CNN"

Some general comments in this regard.

-        The title should not include acronyms, we should make it attractive for broader audiences.

-        About the abstract:

n  Quantitative elements included in the abstract should facilitate comprehension of the manuscript and its possible technical impact. In such sense we recommend some rationalization, comparison with similar valuable results now accepted in the SoA

n  Relation with the restriction included in the title (with CNN) is not clear at all. Why only CNN, why is it the most and guiding technique? What happens with DL?

-        About Introduction:

n  Initial sentences/motivating ones should be supported by quantitative elements, otherwise they could not be considered as technical supports.

n  “None of the existing studies”, instead “as far as these authors known”, is it?

n  The testing countries selected, more important even than the list, which criteria were used for this selection?

-        About Materials and Methods:

n  Complete captions in all the figures. At least a sentence indicating why is the figure at hand relevant to comprehend the manuscript message.

n  Figure 1 supports your work. Even when you should summarize its role in the caption (not done at this stage) you should carefully explain all the elements there included in the main text.

n  Probably them, we could understand why such variable section is there introduced. Otherwise, explain it.

-        About literature review.

n  It is essential to make a review that synthesize how do other experts/papers do contribute with similar research. In this sense, you should mention “areas” and “related contents”.

n  Consider the inclusion of some references to what Artificial Intelligence is doing, for instance:

·       Danilo Cavaliere, Giuseppe Fenza, Vincenzo Loia, Francesco Nota (In Press). "Emotion-Aware Monitoring of Users’ Reaction With a Multi-Perspective Analysis of Long- and Short-Term Topics on Twitter", International Journal of Interactive Multimedia and Artificial Intelligence, vol. In Press, issue In Press, no. In Press, pp. 1-10. https://doi.org/10.9781/ijimai.2023.02.003

·       Chun-Hao Chen, Po-Yeh Chen, Jerry Chun-Wei Lin (2022). "An Ensemble Classifier for Stock Trend Prediction Using Sentence-Level Chinese News Sentiment and Technical Indicators", International Journal of Interactive Multimedia and Artificial Intelligence, vol. 7, issue Special Issue on Artificial Intelligence in Economics, Finance and Business, no. 3, pp. 53-64. https://doi.org/10.9781/ijimai.2022.02.004

·       Mahesh G. Huddar, Sanjeev S. Sannakki, Vijay S. Rajpurohit (2021). "Attention-based Multi-modal Sentiment Analysis and Emotion Detection in Conversation using RNN", International Journal of Interactive Multimedia and Artificial Intelligence, vol. 6, issue Regular Issue, no. 6, pp. 112-121. https://doi.org/10.9781/ijimai.2020.07.004

-        About “Methodology”

n  Linea 115-120 are not clear at all.

n  Lines 140-143 requires more detailed explanation and explicit connection with the manuscript objectives.

n  Interpretation and strict discussion of “opinions” included on table 3 are required.

n  Same for table 4.

n  Reasons behind eq.1?

n  Insets on figures 3-5 are not readable.

The conclusions, in line with such indications already mentioned for the Introductory section, should appeal for communication with broader audiences. In such regards, the use of rationalize relations, instead of absolute numbers is mandatory. What is relevant should be explicitly discussed.

Author Response

First of all, I would like to thank the reviewer for taking an interest in this study. I found all the suggestions valuable, appropriate, and rational. Hence, I have taken keen interest in addressing all the concerns within the updated manuscript. I would like to take this opportunity for thanking you again for your valuable time.

  Re. the manuscript “A New Social Media Driven Electronic-Warfare-Threat Intelligence with CNN"

Some general comments in this regard.

1)      The title should not include acronyms, we should make it attractive for broader audiences.

Many thanks for this suggestion. I agree with this concern. Hence, I have removed acronyms like CNN from the title. Moreover, instead of using “Electronic-Warfare”, I have now used “Cyber” for making it more attractive for the broader audiences. The new title is “A New Social Media Driven Cyber-Threat Intelligence”.

-        About the abstract:

2)  Quantitative elements included in the abstract should facilitate comprehension of the manuscript and its possible technical impact. In such sense we recommend some rationalization, comparison with similar valuable results now accepted in the SoA

Many thanks for this valuable suggestion. Accordingly, I have updated the abstract with “while most of the existing works only works on single language”. Moreover, I have updated the manuscript in several places like line 61, line 415, line 423/424. The outcome of the presented approach is now compared with existing literature, as per the valuable suggestion of the reviewer.

  • Danilo Cavaliere, Giuseppe Fenza, Vincenzo Loia, Francesco Nota (In Press). "Emotion-Aware Monitoring of Users’ Reaction With a Multi-Perspective Analysis of Long- and Short-Term Topics on Twitter", International Journal of Interactive Multimedia and Artificial Intelligence, vol. In Press, issue In Press, no. In Press, pp. 1-10. https://doi.org/10.9781/ijimai.2023.02.003
  • Chun-Hao Chen, Po-Yeh Chen, Jerry Chun-Wei Lin (2022). "An Ensemble Classifier for Stock Trend Prediction Using Sentence-Level Chinese News Sentiment and Technical Indicators", International Journal of Interactive Multimedia and Artificial Intelligence, vol. 7, issue Special Issue on Artificial Intelligence in Economics, Finance and Business, no. 3, pp. 53-64. https://doi.org/10.9781/ijimai.2022.02.004
  • Mahesh G. Huddar, Sanjeev S. Sannakki, Vijay S. Rajpurohit (2021). "Attention-based Multi-modal Sentiment Analysis and Emotion Detection in Conversation using RNN", International Journal of Interactive Multimedia and Artificial Intelligence, vol. 6, issue Regular Issue, no. 6, pp. 112-121. https://doi.org/10.9781/ijimai.2020.07.004

3)  Relation with the restriction included in the title (with CNN) is not clear at all. Why only CNN, why is it the most and guiding technique? What happens with DL?

Many thanks for valuable concern. I find this concern justified and rational. Hence, I have addressed this concern with the following newly added paragraphs:

Within Introduction:

“CNN was chosen as opposed to other, deep learning methods like Recurrent Neural Network (RNN), since CNN have reportedly been used in mobile platform for solving different research problems [21]. RNN, Long Short-Term Memory networks (LSTMs) and other DL implementations are not supported on low-code mobile platforms [21].”

Within Discussion:

“It should be reiterated that CNN was implemented with low-code solutions provided by Microsoft Power Platform [21]. This low-code application of CNN allows seamless mobile deployment of CNN (as shown in Figure 4 and Figure 5) without writing any device specific codes (i.e., targeting iOS or Android). As a result, CNN have been in popularly used in solving deep learning problems in the area of landslides, tornado, global events, and social media analysis as shown in [11] [56] [57] [58] [59] [67] [68] [69] [70].”

-        About Introduction:

  Initial sentences/motivating ones should be supported by quantitative elements, otherwise they could not be considered as technical supports.

Many thanks for this suggestion. Accordingly, initial statements within the introduction sections have been updated. As per the suggestion, quantitative elements like 1 Trillion US dollar in 2020, and 10.5 Trillion US dollar projected damage in 2025. Hence, now the introduction starts with following:

“Cyber-attack and threat caused about 1 trillion US dollar damage to global economy in 2020 [1]. This threat is projected to grow more than ten-folds by 2025 with 10.5 trillion (in USD) damage to global economy [2]”.

4) “None of the existing studies”, instead “as far as these authors known”, is it?

Many thanks for this suggestion. I concur with this recommendation and as a result, I have updated sentence as “According to literature and best of my knowledge, existing studies do not report DL-based analytical capability on global country-wise (i.e., national level) Cyber-threats.”

5)  The testing countries selected, more important even than the list, which criteria were used for this selection?

Many thanks for this observation. Accordingly, I have added the following paragraph to deal with this valuable concern:

“It should be mentioned that these countries were not selected in a random manner. They were selected by considering a range of factors like number of social media users (i.e., US and India having huge number of Twitter Users [14]), concerns of the social media user (i.e., US Tweet users perceiving China and Russia being cyber threat [15] [16]), and current issues (e.g., Russia-Ukraine cyber war, Cyber-attacks in Australia [17] [18] [19] [20] etc.).”

 -        About Materials and Methods:

6)  Complete captions in all the figures. At least a sentence indicating why is the figure at hand relevant to comprehend the manuscript message.

Many thanks for this suggestion. Captions for Figure 1, 2, 3, 4, and 5 have been updated accordingly.

7)  Figure 1 supports your work. Even when you should summarize its role in the caption (not done at this stage) you should carefully explain all the elements there included in the main text.

Probably them, we could understand why such variable section is there introduced. Otherwise, explain it.

Many thanks for valuable suggestion. In fact, anonymous reviewer also appreciated this approach and mentioned, “The methodological approach completed a systematic literature review and by investigating the challenges faced by existing literature, the author proposes an AI, ML, and deep learning (i.e., with Convolutional Neural Network) based solution for creating a threat index.”

As a result, following statement have been added before Fig. 1:

“Thus, the system development framework stipulated by The Open Group Architectural Framework (TOGAF) and Archimate standard were broadly followed [21] as seen from Figure 1.” Moreover, the caption of Figure 1 now explains the intent as “Figure 1. Overview of designing a new Cyber-Index system by systematic literature review, finding challenges, developing requirements, and finally designing solution”.

In fact, the solution development pattern followed in Figure 1 follows Enterprise Architectural frameworks like TOGAF and Archimate. Thus, Figure 1, Table 2, Table 3, and Table 3 are essential part of this study. At the end of Section 2, following sentence have been added:

“This section described how challenges from existing cyber intelligence systems were evaluated (i.e., Table 2), how requirements were developed (i.e., Table 3), and how solutions were built (i.e., Table 4) following Architecture Development Method (ADM) [21].”

-        About literature review.

10)  It is essential to make a review that synthesize how do other experts/papers do contribute with similar research. In this sense, you should mention “areas” and “related contents”.

Many thanks for this suggestion. Instead of 63 references, the updated manuscript contains 78 reference entries. Section 2.1. (i.e., Systematic Literature Review) and Section 2.2 (i.e., Challenges of social media Driven Cyber Solution) highlights existing works as well as the gaps found within existing body of knowledge. Gaps found within the existing body of knowledge is further described from like 100 to line 128. Updated, table 1, also summarizes the gaps of existing works. These gaps are fulfilled by the presented CNN-based solution within the rest of the paper. Lines 249 to line 264 explains different areas of research works solved by CNN based solution. Moreover, we have added the following statement at the end of discussion section, showing areas of research solved with CNN:

“As a result, CNN have been in popularly used in solving deep learning problems in the area of landslides, tornado, global events, and social media analysis as shown in [11] [58] [59] [60] [61] [67] [68] [69] [70]”

11) Consider the inclusion of some references to what Artificial Intelligence is doing, for instance:

  • Danilo Cavaliere, Giuseppe Fenza, Vincenzo Loia, Francesco Nota (In Press). "Emotion-Aware Monitoring of Users’ Reaction With a Multi-Perspective Analysis of Long- and Short-Term Topics on Twitter", International Journal of Interactive Multimedia and Artificial Intelligence, vol. In Press, issue In Press, no. In Press, pp. 1-10. https://doi.org/10.9781/ijimai.2023.02.003
  • Chun-Hao Chen, Po-Yeh Chen, Jerry Chun-Wei Lin (2022). "An Ensemble Classifier for Stock Trend Prediction Using Sentence-Level Chinese News Sentiment and Technical Indicators", International Journal of Interactive Multimedia and Artificial Intelligence, vol. 7, issue Special Issue on Artificial Intelligence in Economics, Finance and Business, no. 3, pp. 53-64. https://doi.org/10.9781/ijimai.2022.02.004
  • Mahesh G. Huddar, Sanjeev S. Sannakki, Vijay S. Rajpurohit (2021). "Attention-based Multi-modal Sentiment Analysis and Emotion Detection in Conversation using RNN", International Journal of Interactive Multimedia and Artificial Intelligence, vol. 6, issue Regular Issue, no. 6, pp. 112-121. https://doi.org/10.9781/ijimai.2020.07.004

Many thanks for valuable suggestion. All the of specified references have been added (please refer to References [14-15]. Moreover, I have added other relevant references. Moreover, these papers have been referred at various places (e.g., line 61, line 415, line 423/424).

-        About “Methodology”

12)  Linea 115-120 are not clear at all.

Many thanks for valuable suggestion. The following statements been added within section 2.2.4:

“These biases may be based on race, religion, culture, ethnicity, or even gender. For example, a person living in US might have a preconception that most of the cyber-attacks are of Russian, Chinese, or North Korean origins. People of Iran, may think that US conducts most of the cyber-attack to create religious tensions within Middle East.”

13)  Lines 140-143 requires more detailed explanation and explicit connection with the manuscript objectives.

Many thanks for valuable suggestion. Within the updated manuscript, I have now made it clear the apparent relation of Table 2, 3, and 4 within the context of the objective of this paper (i.e., to create a new social media based cyber threat intelligence) with the following sentence (at the end of section 2):

“This section described how challenges from existing cyber intelligence systems were evaluated (i.e., Table 2), how requirements were developed (i.e., Table 3), and how solutions were built (i.e., Table 4) following Architecture Development Method (ADM) [14].”

Line 140-143 highlights one of the critical requirement of the presented system (which is “Deeper Understanding with AI”). To further clarify line 140-143 and explicit connection with manuscript objective, I have also added following:

“This paper applies NLP based algorithms to generate deeper understanding of Cyber-threats.”

The first and second paragraph of Section 3. Result & Discussion section talks in details about NLP implementation (e.g., Sentiment Analysis, Named Entity Recognition, Translation) within this paper.

14)  Interpretation and strict discussion of “opinions” included on table 3 are required.

Many thanks for this advice. Indeed, how the requirements have been derived from analyzing the challenges of existing system (following enterprise architectural methodology) needs to be discussed. That’s why we have added the following:

  1. a) “as appears in Table 3" within section 2.3
  2. b) “This section described how challenges from existing cyber intelligence systems were evaluated (i.e., Table 2), how requirements were developed (i.e., Table 3), and how solutions were built (i.e., Table 4) following Architecture Development Method (ADM) [14].” at the end of section 2.

15)  Same for table 4.

Many thanks for this advice. Indeed, how the solutions have been derived from analyzing the requirements developed in Table 3 (following enterprise architectural methodology) needs to be discussed. That’s why we have added the following:

“This section described how challenges from existing cyber intelligence systems were evaluated (i.e., Table 2), how requirements were developed (i.e., Table 3), and how solutions were built (i.e., Table 4) following Architecture Development Method (ADM) [14].” at the end of section 2.

16)  Reasons behind eq.1?

Many thanks for this suggestion. It has been indicated now (in line 61), that “number of social media user” is a critical factor for this research. Since the number of social media varies greatly from country to country, the country-wise indexes have different scales for different countries. Therefore, eq. 1 is required to normalize all the indexes pertaining to different countries under same scale. Following statement has been added to explain this fact:

“For all these indexes, the minimum and maximum values could be significantly different since the numbers of social media posts varies greatly from one country to another. Thus, Eq. 1 generates all the country-wide indexes under same scale.”

17)  Insets on figures 3-5 are not readable.

Many thanks for this valuable concern. Captions of Figure 3 to 5 have been updated to clarify that these images the device screenshots from Windows, Android, and iOS deployment. Following new statements (within Section 3) have also been added:

“While Figure 4 shows deployment through Power BI Mobile App on Android environment, Figure 5 depicts deployment within iOS environment. Since both Figure 4 and Figure 5 shows deployment within different mobile platforms, the contents might be difficult for viewing. Hence, a publicly accessible web link is available at https://app.powerbi.com/view?r=eyJrIjoiYWJjOGY5YTUtZDBlNy00MTg1LWFkMTMtM2RmYzYzODQ1NzE1IiwidCI6IjBkMWI4YmRlLWZmYzEtNGY1Yy05NjAwLTJhNzUzZGFjYmEwNSJ9&pageName=ReportSection”.

The readers can now access this newly created URL (publicly accessible) to comprehend the contents of Fig. 3 to 5. 

18) The conclusions, in line with such indications already mentioned for the Introductory section, should appeal for communication with broader audiences. In such regards, the use of rationalize relations, instead of absolute numbers is mandatory. What is relevant should be explicitly discussed.

Many thanks for valuable suggestion. Accordingly, I have updated the conclusion. I have rationalized the outcomes relating them existing studies (e.g., line 415, line 423/424). This will connect with broader audiences. Moreover, I have also added the following paragraph within the conclusion section:

“However, since the methodology described within this study explicitly relies on Social media data for creating the index (and normalizing the indexes with Eq. 1), any threats to social media platforms or infrastructure would be a threat to the presented system. For example, if Twitter platform is under cyber-attack, this presented system would fail. In another example, if a group of people or organization conduct information operation by generating a huge number of fake posts (i.e., related to cyber), then that would also be a threat to the presented system.”

Round 2

Reviewer 1 Report

Thank you so much for addressing all my comments. I do not have further comments.

Reviewer 2 Report

Accept as is